# Health Needs Assessment of Five Pennsylvania Plain Populations

**DOI:** 10.3390/ijerph16132378

**Published:** 2019-07-04

**Authors:** Kirk Miller, Berwood Yost, Christina Abbott, Scottie Thompson Buckland, Emily Dlugi, Zachary Adams, Varun Rajagopalan, Meryl Schulman, Kimberly Hilfrank, Mara A. Cohen

**Affiliations:** 1Department of Biology, Franklin & Marshall College, 415 Harrisburg Ave., Lancaster, PA 17603, USA; 2Floyd Institute Center for Opinion Research, Franklin & Marshall College, Lancaster, PA 17603, USA; 3Department of Psychology, Franklin & Marshall College, Lancaster, PA 17603, USA; 4Johns Hopkins Hospital, Baltimore, MD 21287, USA; 5U.S. News & World Report, Washington, DC 20007, USA; 6Ernst and Young Advisory Services, New York, NY 10036, USA; 7Center for Health Care Strategies, Trenton, NJ 08619, USA; 8School of Public Health, The University of Michigan, Ann Arbor, MI 48109, USA

**Keywords:** Amish, Old Order Mennonite, health needs, household survey

## Abstract

We performed a health needs assessment for five Plain communities in Pennsylvania from a random sample of households, comparing them to the general population of Pennsylvania adults. Plain respondents were more likely to drink well water, as likely to eat fruit and vegetables and much more likely to drink raw milk and be exposed to agricultural chemicals. Plain respondents were less likely to receive screening exams compared to the general population and there was variation from settlement to settlement in whether respondents had a regular doctor, whether they received preventive screenings or had their children vaccinated, with Mifflin County Amish generally lowest in these and Plain Mennonites highest. Plain respondents reported good physical and mental health compared to the general population but Groffdale Mennonite respondents had a high proportion of diagnoses of depression and were more likely to be receiving treatment for a mental health condition. Most Plain respondents would want a spouse tested for genetic disease with Mifflin County Amish least in favor of these tests. Despite their geographic and genetic isolation, the health of Plain communities in Pennsylvania is similar to that of other adults in the state.

## 1. Introduction

Pennsylvania is home to a number of communities that isolate themselves from the larger society and that are descended from Anabaptist sects that migrated to America between about 1683 and 1860 to avoid religious persecution in Europe [1]. Lancaster County, in the southeastern part of the state, is the center of distribution of these Amish and Mennonite groups that, partly due to shortages of farmland and partly from a desire to isolate themselves, have spread throughout Pennsylvania, much of the rest of the United States and Canada, Mexico, the Caribbean and Central America [2]. These groups do not have health care providers of their own because of restrictions on education but they do make use of modern medicine and health care providers from outside their communities.

Because of their isolation, the health needs of these communities are often overlooked in assessments of large geographic areas in the service region of hospitals. Because they do not assimilate, have rigid rules about the use of technology and often do not receive formal education beyond the eighth grade, the demographic characteristics of these communities are often very different from those of the larger society.

The Old Order Amish and Old Order Mennonite groups that eschew many of the trappings of modern life call themselves “plain.” Because these Plain groups were all founded by small numbers of families, they carry high numbers of certain disease-causing genes (and small numbers of others). Researchers have identified over 100 disease-causing mutations in the Plain communities [3] but life expectancy and infant mortality in these communities are unknown.

We have previously reported on a health needs assessment of the Amish and Plain Mennonites of Lancaster County [4]. Here we report a health needs assessment of two isolated and relatively conservative Pennsylvania Amish communities and compare them directly to the Amish and Mennonites of Lancaster County and, when possible, to the general population of Pennsylvania.

Amish families founded a settlement in Somerset County, PA, in the western part of the state, perhaps as early as 1767 [5]. While this is the second oldest extant Amish community, it is small and, uniquely, Sunday services are held in meeting houses rather than homes. Members migrated west to found the now very large Amish settlements in Holmes County, Ohio and Elkhart and Lagrange Counties, Indiana [5]. The Somerset County settlement has an unusually high carrier frequency of a mutation in MTHFR that can lead to severe cerebral atrophy [3].

Amish families settled in Mifflin County, PA starting in 1791 [6]. This region, called the Big Valley, includes at least 12 groups that are Amish or descendants of Amish groups [6,7]. This settlement has a high prevalence of propionic acidemia, which can lead to seizures and severe heart problems and of a mutation in TSPYL that can lead to sudden infant death [3,8].

There is little evidence-based research on Plain people’s health, health care and preventive care and behaviors and exposures that affect health. This research provides data on some of these important health indicators.

## 2. Materials and Methods

The Center for Opinion Research at Franklin & Marshall College conducted a random sample household survey by mail between August 2014 and May 2015 that assessed the health needs of adult Old Order Amish and Old Order Mennonite individuals (18 years of age and older) living in five settlements in Pennsylvania.

The purpose of the survey was to gain an understanding of the current health and health needs of these Plain communities, to assess differences between settlements and to measure how their perceptions of modern medicine and technology may be altering the Plain way of life. The study was approved by the Institutional Review Board of Franklin & Marshall College (#354 approved 2/28/14).

### 2.1. Populations and Sample

The Address Book of the Lancaster County Amish [9], Directory of the Groffdale Conference Mennonite Churches [10], Directory of the Weaverland Conference Mennonite Churches [11], Old Order Amish Directory of Somerset County, PA Letard, West Virginia [12], Old Order Amish Directory of Mifflin and Juniata Counties [13] and Nebraska Amish Directory [14] were used as the sampling frames. These contained the most complete available listing of households in their respective groups. Estimated population sizes, sample sizes and response rates are given in Table 1. For the purposes of this analysis the groups of Mifflin County Amish were combined.

### 2.2. Survey Methods and Measures

Survey methods and measures have been described elsewhere [4] (a copy of the survey is available from the corresponding author). The response rate averaged 49% and varied from settlement to settlement (Table 1).

We report here the proportion of respondents for each survey question for the five surveyed settlements. Where possible, we also report the proportion answering similarly from surveys of Pennsylvania residents. We compare responses from Plain settlements using Pearson Chi-square or analysis of variance.

## 3. Results

The 682 Plain respondents ranged in age from 18 to 91, mean 46.4 years. Plain respondents were more likely to be married and to have large families than the general population of adults living in Pennsylvania (Table 2). Plain respondents were more likely to live on a farm and to drink water from a private well.

Plain respondents were as likely to eat fruit and vegetables as the general population of adults in Pennsylvania (Table 3) and probably much more likely to drink raw milk. There was variation among settlements in these measures with Mifflin County Amish less likely to eat fruit and vegetables and Weaverland Mennonites less likely to drink raw milk. Plain respondents were also highly likely to report being exposed to agricultural chemicals. There was considerable variation from settlement to settlement in whether respondents had a regular doctor, whether they availed themselves of preventive screening and whether they had their children vaccinated, with the Mifflin County Amish generally the lowest in these measures and the Plain Mennonite groups the highest. Plain respondents were generally less likely to receive screening exams compared to the general population of Pennsylvania. Plain respondents had a Body–Mass Index (BMI: weight kg/height m^2^) comparable to the adult population of Pennsylvania although there was variation among settlements, Somerset County Amish having higher and Lancaster County Amish and Groffdale Mennonites having lower proportions of overweight and obese BMI. Plain respondents almost universally received prenatal care.

Plain respondents report good physical health and generally fewer diagnosed health conditions compared to the general population of Pennsylvania adults (Table 4). Notably, Plain groups report fewer diagnoses of asthma and hypertension compared to adults in Pennsylvania and many diagnoses of urinary tract infections, vaginal yeast infections and thyroid problems. Plain respondents also report fewer diagnoses of obesity and arthritis and fewer pregnancy complications. Plain Mennonite groups generally report more diagnosed conditions compared with Amish groups and strikingly higher diagnoses of cancer.

About 32% of Plain respondents needed medical care in the past 12 months. Of these, the most common answers to questions about why care might be delayed were because of expense (30% of respondents) and because of uncertainty about where to go for care (16% of respondents).

Plain respondents generally report better mental health compared with the general population of adults living in Pennsylvania. Very few scored high on measures of depression and bipolar disorder and many scored very high on measures of life satisfaction and social support (Table 5). Of 620 Plain respondents, 79 (12.7%) had one or more poor mental health days, compared with 35% of adults in Pennsylvania. Groffdale Mennonite respondents have a high proportion of diagnoses of depression, have a high number of poor mental health days and are more likely to be receiving treatment for a mental health condition but are also likely to go to a doctor if they felt they had a mental health problem. Despite having a high number of poor mental health days, they receive the social support they need and are very satisfied with their life.

Respondents’ attitudes towards genetic testing and fatalism are summarized in Table 6. Although there was variation among settlements, with Mifflin County Amish least in favor, most respondents would want a spouse tested for genetic disease. Fewer respondents would want an unborn child tested for genetic disease and, again, Mifflin County Amish were least in favor. Mifflin County Amish babies were also least likely to have received newborn genetic screening. There was broad agreement among respondents on questions about fatalism with only very small proportions agreeing that their health was determined by fate (Table 6) and only about one-fourth agreeing that following medical advice would not affect the likelihood of a serious disease.

The relationships between respondents’ views on fatalism and genetic testing are in Table 7. There does not appear to be a relationship between views on fatalism and the desire to test a spouse or unborn child for genetic disease.

Fifty-one respondents had someone in their family who is a patient at the Clinic for Special Children. There were no differences between those who had or did not have a family member a patient at the Clinic on answers to questions about fatalism (*p* = 0.34 and *p* = 0.06), nor in their desire to test an unborn child for genetic disease (*p* = 0.80). Those with a family member a patient at the Clinic were slightly less likely to want a spouse tested for genetic disease (76% vs. 87%, *p* = 0.02).

## 4. Discussion

There was considerable variation from settlement to settlement in health and health behaviors. In general, Mennonite groups were most likely to receive preventive screening and most likely to have received diagnoses of disease, and Mifflin County Amish were least likely to receive screenings and least likely to have received diagnoses of disease. Among Plain respondents, Groffdale Mennonite respondents have a high proportion of diagnoses of depression and are more likely to be receiving treatment for a mental health condition. Groffdale and Weaverland Mennonites report higher diagnoses of cancer. Differences between groups may reflect differences in diagnosis rather than different underlying disease prevalence.

Few Plain respondents seem to have attitudes that suggest fatalism, and most are in favor of genetic testing. With little variation between settlements, only about one-fourth of respondents agree that “If someone is meant to have a serious disease, it doesn’t matter what doctors and nurses tell them to do, they will get the disease anyway.” and only one-tenth agree that “My health is determined by fate.” Even among those who agree with the statements, at least four-fifths would want a spouse tested for genetic disease and about half would want an unborn child tested. Mifflin County Amish, the most geographically isolated settlement, scored highest on measures of fatalism and lowest on the desire for genetic testing.

Despite some variation between Plain groups, the health of members of Old Order groups appears remarkably similar to members of the general population of Pennsylvania, despite living a very different lifestyle and being genetically isolated from them. There are some notable differences, though, between Plain respondents and Pennsylvania adults. Plain respondents are more likely to live on a farm, drink water from a private well and be exposed to agricultural chemicals than other Pennsylvania adults; they have a high prevalence of certain conditions (notably, anemia, thyroid problems and urinary tract and vaginal yeast infections) and lower prevalence of others (notably, depression and asthma). Plain respondents were less likely than members of the general population to receive preventive screening, which may reflect attitudes towards healthcare or its difficulty and expense.

Asthma prevalence may be lower in Plain communities [15,16]. The Lancaster County Amish community also has lower rates of low birth weight compared with women in Central Pennsylvania generally [17].

Plain communities may also have different numbers of diagnoses of various mental illnesses compared with the general population. Fuchs et al. [18] report higher prevalence of depression among the Amish while Miller et al. [17] reports lower. Bipolar disorders may be more prevalent among the Amish and some plausibly causative pathogenic alleles have been identified [19,20]. Differences in the prevalence of mental health conditions between Plain people and the general population have been of interest to investigators both because of social influences on mental health and to isolate genetic influences on mental health.

This work shows, we think, that surveying Plain communities by mail is feasible. Thus, knowledge of other, more isolated and poorer, Plain communities might be easily and relatively cheaply obtained. Furthermore, this work shows that, despite their relative geographic and genetic isolation and with a few notable exceptions, the health of adult members of the Plain communities in Pennsylvania is similar to that of other adults in the state.

The desire of Plain people to farm, combined with the expense of farm land, will continue to cause migration to less populated areas as Plain populations continue to grow. Thus, contact between Plain communities and other parts of the general population and particularly with local medical care providers will continue to increase. Providers will be particularly interested in those areas where Plain people’s health differs from their neighbors. Why do Plain families apparently have fewer low birth weight babies and less asthma and depression, for example? What are the public health and medical care implications of the high prevalence of maple syrup urine disease, glutaric aciduria type I and other genetic disease in these communities? What are the implications for urgent care and chronic disease management? Future work should explore these questions and survey other, smaller, poorer, more isolated Plain groups to better understand the variation that exists within and between Plain populations.

## 5. Conclusions

A health needs assessment of five Pennsylvania Plain communities was performed on a random sample of households. There was variation from settlement to settlement in whether respondents had a regular doctor, whether they received preventive screenings or had their children vaccinated, with more conservative groups generally lowest in these and least conservative groups highest. Plain respondents reported good physical and mental health compared to the general population. Despite their geographic and genetic isolation, the health of Plain communities in Pennsylvania is similar to that of other adults in the state.

## Figures and Tables

**Table 1 ijerph-16-02378-t001:** Population estimates, sample sizes and response rates for five surveyed Plain communities.

Settlement	Population	Sample	Response
Lancaster County Amish	35,070 ^a^	458	49%
Groffdale Mennonite	4950 ^b^	149	63%
Weaverland Mennonite	4767 ^c^	114	60%
Mifflin County Amish	3120 ^d^	489	31%
Somerset County Amish	700 ^e^	183	83%

^a^ 2016 Young Center for Anabaptist and Pietist Studies. Available online: http://groups.etown.edu/amishstudies/ (accessed 21 June 2017). ^b^ 2002 Lancaster County total population Kraybill and Hurd Horse-and-Buggy Mennonites p. 3. ^c^ Adult members 1994 Landis, Ira D. and Richard D. Thiessen. “Weaverland Mennonite Conference.” Global Anabaptist Mennonite Encyclopedia Online. October 2010. Available online: http://gameo.org/index.php?title=Weaverland_Mennonite_Conference&oldid=135260 (accessed 21 June 2017). ^d^ 2012 Kraybill, Johnson-Weaver and Nolt, The Amish p. 186. ^e^ Yoder, Samuel L. “Meyersdale-Springs Old Order Amish Settlement (Somerset County, Pennsylvania, USA).” Global Anabaptist Mennonite Encyclopedia Online. 1990. Available online: http://gameo.org/index.php?title=Meyersdale-Springs_Old_Order_Amish_Settlement(Somerset_County,_Pennsylvania,_USA)&oldid=113519 (accessed 2 June 2017).

**Table 2 ijerph-16-02378-t002:** Demographic characteristics of 5 Pennsylvania Plain communities and the general Pennsylvania population.

	Settlement					General Population
	Lancaster County Amish	Groffdale Mennonite	Weaverland Mennonite	Mifflin Country Amish	Somerset Country Amish	
Average age of respondent	46	47.1	50	46.3	44.8	48 ^a^
Married	84%	66%	85%	88%	84%	46% ^b^
Average number of children of respondent	5.1	3.9	4.7	5.2	5.1	1.85 ^c^
Live on a farm	52%	47%	29%	45%	87%	0.2% ^d^
Home built before 1975	49%	60%	55%	50%	71%	72% ^e^
Drinking water from a private well	95%	96%	92%	87%	100%	20% ^f^

^a^ Median age of PA residents 18 and older US Census 2010. ^b^ Married PA residents 15 and older 2013 PA State Data Center. ^c^ Average number of children per PA family with children US Census 2000. ^d^ PA housing units on farms US Census 2010. ^e^ PA houses built before 1979 American Community Survey 2015. ^f^ Private water wells in PA Penn State Extension private water systems faqs.

**Table 3 ijerph-16-02378-t003:** Behaviors and exposures of 5 Pennsylvania Plain communities and the general Pennsylvania population.

	Settlement					General Population PA
	Lancaster County Amish	Groffdale Mennonite	Weaverland Mennonite	Mifflin Country Amish	Somerset Country Amish	
In a typical week how often do you eat (% responding once a day or more)						
Fruit *	63%	80%	72%	38%	80%	64% ^a^
Vegetables *	55%	82%	68%	31%	71%	76% ^a^
Salad *	26%	6%	6%	9%	16%	
Raw milk *	71%	71%	43%	74%	90%	
Exposed to agricultural chemicals	50%	65%	50%	49%	64%	
Has a regular doctor or health professional *	74%	93%	99%	40%	58%	87% ^b^
Received the following health services in the past 12 months						
Blood cholesterol check *	20%	18%	38%	13%	25%	79% ^c^
Physical checkup *	20%	31%	43%	15%	16%	86% ^d^
Blood pressure test *	43%	54%	72%	30%	46%	
Test for diabetes *	12%	21%	28%	8%	16%	58% ^e^
Flu shot *	3%	11%	25%	2%	0%	43% ^b^
Test for bacterial vaginosis	2%	6%	2%	1%	0%	
Dental exam *	39%	65%	76%	5%	15%	67% ^f^
Pelvic exam *	4%	14%	12%	2%	4%	
Pap smear	7%	16%	21%	2%	3%	69% ^e^
Physical breast exam *	7%	11%	24%	2%	3%	56% ^g^
Mammogram *	2%	8%	19%	0	1%	57% ^g^
Prostate exam	3%	2%	3%	2%	5%	60% ^h^
If you have children, have they been vaccinated? (% responding yes)	52%	86%	95%	25%	41%	79% ^i^
During your or your wife’s last pregnancy, did she visit a midwife or other health professional? (% yes)	100%	100%	100%	98%	99%	72% ^j^
BMI (% overweight or obese) *	50%	58%	69%	62%	71%	66% ^g^

* Significant differences between settlements. ^a^ CDC State indicator report on fruits and vegetables 2013. ^b^ PA Dept. Health 2015 Behavioral health risks of Pennsylvania adults. ^c^ Past 5 years, PA Dept. Health 2015 Behavioral health risks of Pennsylvania adults. ^d^ Past 2 years PA Dept. Health BRFSS/EDDIE. ^e^ Past 3 years PA Dept. Health BRFSS/EDDIE. ^f^ PA Dept. Health Healthy People 2020. ^g^ PA Dept. Health BRFSS/EDDIE. ^h^ PSA test PA Dept. Health BRFSS/EDDIE. ^i^ Fully immunized children 19-35 months PA Dept. Health Healthy People 2020. ^j^ Prenatal care in first trimester PA Dept. Health Healthy People 2020.

**Table 4 ijerph-16-02378-t004:** Health and health conditions of 5 Pennsylvania Plain communities and the general. Pennsylvania population.

	Settlement					General Population PA
	Lancaster County Amish	Groffdale Mennonite	Weaverland Mennonite	Mifflin County Amish	Somerset County Amish	
In general, would you say your health is…						
Self-reported health fair or poor	13%	10%	6%	15%	21%	16% ^a^
Has a doctor ever told you that you have…						
Asthma	6%	3%	6%	6%	5%	15% ^a^
Hypertension *	6%	22%	28%	10%	16%	33% ^a^
High cholesterol *	17%	22%	30%	9%	17%	36% ^a^
Coronary disease	4%	3%	9%	3%	6%	6% ^a^
Stroke	1%	2%	2%	4%	2%	5% ^a^
Blood clot	5%	11%	3%	7%	5%	
Epilepsy	1%	0%	3%	2%	0%	
COPD	3%	0%	3%	1%	3%	7% ^a^
Obesity	14%	10%	18%	7%	12%	30% ^a^
Anemia	21%	30%	35%	18%	26%	
Anxiety or depression *	8%	21%	12%	10%	6%	18% ^a^
Arthritis	14%	16%	18%	15%	11%	29% ^a^
Thyroid problems	16%	10%	14%	13%	12%	
Chlamydia	1%	0%	0%	0%	0%	0.6% ^a^
Herpes	1%	1%	2%	1%	1%	
Gonorrhea	0%	0%	0%	0%	0%	0.1% ^a^
Syphilis	0%	0%	0%	0%	0%	0.006% ^a^
Diabetes *	2%	3%	12%	4%	2%	10% ^a^
Cervical cancer	0%	1%	0%	0%	1%	0.02% ^a^
Urinary tract infection *	13%	20%	33%	16%	22%	
Endometriosis	1%	6%	3%	2%	2%	
Bacterial vaginosis	0%	3%	3%	1%	0%	
Vaginal yeast infection	17%	22%	27%	12%	20%	
Pelvic inflammatory disease	1%	1%	0%	0%	0%	
Pregnancy complications	2%	6%	8%	3%	8%	33.5% ^b^
Cancer *	2%	7%	12%	2%	5%	1.3% ^a^

* Significant differences between settlements. ^a^ PA Dept. Health BRFSS/EDDIE. ^b^ PA Dept. Health Healthy People 2020.

**Table 5 ijerph-16-02378-t005:** Mental health and social support in 5 Pennsylvania Plain communities and the general Pennsylvania population.

	Settlement					General Population PA
	Lancaster County Amish	Groffdale Mennonite	Weaverland Mennonite	Mifflin County Amish	Somerset County Amish	
Has a doctor ever told you that you have anxiety, depression or bipolar disorder *	8%	21%	12%	10%	6%	20% ^a^
In past 4 weeks have you accomplished less than you wanted because of emotional problems, % often or all the time	1%	1%	2%	0	2%	
One or more poor mental health days	10%	18%	12%	12%	15%	35% ^a^
Taking medication or receiving treatment for a mental health condition *	7%	20%	10%	6%	2%	
CESD depression score, % scoring 4–5 on a scale of 0–5 ^b^	2%	0	0	3%	3%	
Mood disorder score, % scoring 3 on a scale of 0–3 ^c^	3%	4%	5%	3%	5%	
Would go to a doctor if felt had a mental problem, % very likely or likely *	58%	80%	83%	58%	38%	
PHQ-8 depression scale, % moderate or severe symptoms ^d^	1%	1%	0	2%	5%	
SOCIAL SUPPORT						
How often do you get the social and emotional support you need? % usually or always	87%	90%	91%	81%	91%	92% ^e^
How satisfied are you with your life? % satisfied or very satisfied *	80%	86%	94%	81%	65%	94% _e_
% responding often or all the time on 3 measures of social support ^f^	88%	88%	94%	91%	92%	

* Significant differences between settlements. ^a^ PA Dept. Health BRFSS/EDDIE 2014. ^b^ 5 questions from the Center for Epidemiologic Studies Depression Scale. ^c^. 3 questions from the Mood Disorder Questionnaire screening for bipolar disorder. ^d^ Personal Health Questionnaire Depression Scale. ^e^ PA Dept. Health BRFSS/EDDIE 2010. ^f^ Medical Outcomes Study Social Support Survey.

**Table 6 ijerph-16-02378-t006:** Attitudes towards genetic testing and fatalism in 5 Pennsylvania Plain communities.

	Settlement				
	Lancaster County Amish	Groffdale Mennonite	Weaverland Mennonite	Mifflin County Amish	Somerset County Amish
If you found out you were a carrier for a genetic disease, would you want your spouse tested for it? (% yes) *	85%	88%	90%	71%	95%
Would you want to know if your child was going to be affected by a particular genetic disease before he or she was born if a test could tell you this? (% yes) *	46%	56%	65%	27%	47%
After your baby was born, did he or she receive a newborn screening test? (% yes) *	53%	52%	73%	28%	53%
FATALISM					
If someone is meant to have a serious disease, it doesn’t matter what doctors and nurses tell them to do, they will get the disease anyway. (% strongly agree or agree) ^a^	25%	20%	26%	33%	23%
My health is determined by fate. (% strongly agree or agree) *	9%	5%	6%	13%	3%

* Significant differences between settlements. ^a^ 2 questions from a 20 question Fatalism Scale.

**Table 7 ijerph-16-02378-t007:** The relationship between attitudes on genetic testing and fatalism in 5 Pennsylvania Plain communities.

	**If Someone is Meant to Have a Serious Disease, It doesn’t Matter What Doctors and Nurses Tell Them to do, They will Get the Disease**			
		**Strongly Agree or Agree**	**Disagree or Neutral**	
If you found out you were a carrier for a genetic disease, would you want your spouse tested for it?	yes	108 (81%)	341 (88%)	*p* = 0.03
	no	26 (19%)	46 (12%)	
Would you want to know if your child was going to be affected by a particular genetic disease before he or she was born if a test could tell you this?	yes	57 (42%)	178 (48%)	*p* = 0.23
	no	78 (58%)	191 (52%)	
	**My Health is Determined by Fate**			
		**Strongly Agree or Agree**	**Disagree or Neutral**	
If you found out that you were a carrier for a genetic disease, would you want your spouse to be tested for it?	yes	32 (87%)	384 (87%)	*p* = 1
	no	5 (14%)	60 (14%)	
Would you want to know if your child was going to be affected with a particular genetic disease before he or she was born if a test could tell you this?	yes	20 (57%)	206 (48%)	*p* = 0.29
	no	15 (43%)	225 (52%)	

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
