# Peer review of "Health Needs Assessment of Five Pennsylvania Plain Populations"

_ijerph, 2019, doi:10.3390/ijerph16132378_

Round 1
Reviewer 1 Report
Thank you for the opportunity to review the article “Health needs assessment of five Pennsylvania Plain populations” for Int. J. Environ. Res. Public Health. This paper conducted a health needs assessment of adults from five Plain communities in Pennsylvania from a random sample of households, comparing them to each other and the general population of Pennsylvania adults. Below are some comments for the authors to consider:
2.2 Survey Methods and Measures
Please provide once sentence about the method of the survey, was it in person, via mail, internet-based. I see you mentioned via mail-in conclusion, but still, it would be better to provide very brief info about the survey method.
3. Results
Mean age is in the table but not reported in the text. I think it would be also appropriate to report the mean age, in addition to the min and max age of the respondents.
Page 3 line 39. I am not sure the first line of reporting is reflective off the results. Clearly, Mifflin County Amish were twice less likely to eat fruits and vegetable than the general population. I would recommend authors report results at a more granular level for important outcomes to reflect the significant variation within the Plain adults and between the former and the Pennsylvania adult population. Maybe it would be better to report how many Plain communities/settlements out of total (and maybe list them in the brackets with respective percentages for the indicators) had comparable rates (or significantly lower or higher) to the Pennsylvania adult population.
Given the more similarities between Plain groups and the rural residents among the general population, it would be appropriate to compare the health outcomes and behaviors of the Plain groups to the rural Pennsylvania general population instead of the overall general population which includes both rural and urban.
Page 3, line 41. It is not clear from the authors’ reporting how exposure to the agricultural chemical was measured. Other outcomes are more or less clear, but not this one. Please provide more information how it was defined either in the method section or in the table where the results are reported.
Page 4, line 1. I don’t agree that the Plain groups had similar BMI. Again, there is significant variation within the Plain group and between the former and the Pennsylvania population and should be reported as such. For example, authors can say something like: Three Plain groups (list the names) had comparable BMI (BMI percentages for the groups vs. % in Penn. general population) while two groups (list names) had much lower obesity/overweight rate (50% and 58% respectively compared to 66% in general population), etc.
While reporting results in table 3, authors never mentioned the striking difference of having cancer among adults from Plain communities that had between 4 to 10 times higher rates compared to the Pennsylvania general population. I think this should be reported and appropriate implications should be mentioned.
In reporting results from table 4 about the reason for delayed care, I was wondering whether having health insurance coverage was ever asked in the survey. It is common sense to ask this question during health assessment and wondered if authors have asked this question. Also, I wondered if a question about the education level was asked.
Table 6
I wondered if the question in the 4th row should read “newborn genetic screening test”instead of a “newborn screening test”. Also, lower rates in response to this question might indicate that hospital, especially in rural areas, might not have resources to do the test.
Page 7, line 11-13. This must be comments from one of the co-authors. Anyways, it does not feel to belong here.
Discussion
Again, to me, one striking difference that is omitted in this analysis was high rates of cancer in Plain adults, and this should be highlighted and discussed appropriately.
Overall, a more granular reporting of the results for within Plain group differences and differences between the former and the general population, as mentioned in one example above, would be well warranted.
Author Response
response to reviewer #1 IJERPH-537444:
Thank you for your thorough review of our paper, Health needs assessment of five Pennsylvania Plain populations.
We have added text to clarify that ours was a mailed survey p. 2, line 28.
We have added the mean age of respondents p. 3, line 16.
We have added text to emphasize important differences between settlements p. 3, lines 41-43.
We have clarified that exposure to agricultural chemicals is by self-report p. 3, line 44.
We have added text to emphasize important differences between settlements in BMI p. 4, lines 4-6.
We have emphasized the number of cancer diagnoses p. 4, line 56.
The Plain people do not carry health insurance and do not educate past the eighth grade. Other work we have done indicates they do often delay care because of the expense.
The question in the 4th row of Table 6 is the question that was asked in the survey. Feedback we received from Plain individuals asked to comment on a draft survey indicated that omitting the word “genetic” was clearer.
Almost all Plain births are at home and attended by a midwife. Some of these midwives do not take newborn blood spots for genetic screening. Variation among settlements in this probably indicates differences in midwife education and education.
The inappropriate lines on p. 7 have been removed.
A sentence was added to the Discussion highlighting differences in cancer diagnoses p. 7, lines 18-19.
We are developing a statistical model to predict health conditions and uptake of care from demographic characteristics, especially gender and age, and settlement membership. This will allow a much more granular analysis of differences among settlements and their causes.
It is also our intention to develop a comparison of Plain responses to those of other rural Pennsylvanians. The comparison information will need to be gathered county-by-county.
Authors: Your research identifies important needs assessment results for the Plain communities. The study may advance research for targeting this population and tailoring the health promotion and disease management education. You organized your paper well. One suggestion would be more statistical analyses.
Author Response
response to reviewer #2 IJERPH-537444:
Thank you for your review of our paper, Health needs assessment of five Pennsylvania Plain populations.
We are developing a statistical model to predict health conditions and uptake of care from demographic characteristics, especially gender and age, and settlement membership. This will allow a much more granular analysis of differences among settlements and their causes.
Reviewer 3 Report
1. This manuscript represents an observational study describing a unique population viv-a-vis the general health of the contiguous population. It is designed to yield questions for further study.
2. Lines11-13 on page 7 appear to be instructions for a section heading and should be deleted.
3. In the discussion please speculate on the disparity between preventative health practices for Plain people and the general population, and similarities in general health. Do the authors identify any limitations in available health outcomes measures?
4. Please also speculate on what the data indicates about the use of urgent healthcare resources and implications for chronic disease management.
Author Response
response to reviewer #3 IJERPH-537444:
Thank you for your review of our paper, Health needs assessment of five Pennsylvania Plain populations.
The inappropriate lines on p. 7 have been removed.
We have added text to speculate on why Plain people receive less preventive care p. 8, line 7-8.
All our data are self-reports, we do not know life expectancy or infant mortality in Plain communities, so we really don’t have any information on outcomes.
We have added text to highlight the (unknown) implications for urgent care and chronic disease management p. 8, line 31-32.
These are terrific questions that our data only allow us to ask.
We are developing a statistical model to predict health conditions and uptake of care from demographic characteristics, especially gender and age, and settlement membership. This will allow a much more granular analysis of differences among settlements and, perhaps, their causes.